# BiDST: Dynamic Sparse Training is a Bi-Level Optimization Problem

## Abstract

Dynamic Sparse Training (DST) is an effective approach for addressing the substantial training resource requirements posed by the ever-increasing size of the Deep Neural Networks (DNNs). Characterized by its dynamic "train-prune-grow" schedule during training, DST implicitly develops a bi-level structure for training the weights while discovering a subnetwork topology. However, such a structure is consistently overlooked by the current DST algorithms for further optimization opportunities, and these algorithms, on the other hand, solely optimize the weights while determining masks heuristically. In this paper, we extensively study DST algorithms and argue that the training scheme of DST naturally forms a bi-level problem in which the updating of weight and mask is interdependent. Based on this observation, we introduce a novel efficient training framework called BiDST, which for the first time, introduces bi-level optimization methodology into dynamic sparse training domain. Unlike traditional partial-heuristic DST schemes, which suffer from sub-optimal search efficiency for masks and miss the opportunity to fully explore the topological space of neural networks, BiDST excels at discovering excellent sparse patterns by optimizing mask and weight simultaneously, resulting in maximum $2.62\%$ higher accuracy, $2.1\times$ faster execution speed, and $25\times$ reduced overhead. Code will be released.

## 1 Introduction

Deep neural networks (DNNs) have transformed traditional machine learning technology into an intelligent and scalable ecosystem, fueling a wide range of challenging tasks. The arrival of the Artificial General Intelligence (AGI) era coupled with extensive data has catalyzed the development of larger model sizes and an exponential increase in computational requirements, resulting in a great challenge for end-users when implementing edge intelligence locally.

A straightforward yet effective solution is to introduce *sparsity* into DNN training (Deng et al., 2020). As pointed out in Bellec et al. (2018) and Mocanu et al. (2018), weight sparsity can be leveraged to potentially satisfy memory bound along with training computation reduction, with notably higher accuracy compared with directly training a shrinked-size DNN model. In order to obtain such merits, a sparse mask (i.e., a set of binary values that has the same shape as weights and contains only 0 or 1) needs to be developed, in which the 1s activate a portion of the original dense network by selecting weights (i.e., usually less than 30% of the total weights) for training and inference. Under such circumstances, finding the binary mask before or during training surges as the key enabler to provide accurate and efficient learning dynamics (Lee et al., 2019; Wang et al., 2020b; Mocanu et al., 2018). Among a variety of methods, the Dynamic Sparse Training (DST) (Mocanu et al., 2018) has emerged with the ability to optimize weights and find the mask concurrently, and rapidly becomes the prevailing approach owing to its high accuracy, efficiency, and ease of implementation.

In prior DST algorithms, a sparse mask keeps the entire training phase following an "always sparse" regime, and is evolving its topology by iteratively pruning weights with smallest magnitude and then growing other weights back. As shown in Figure 1 (a), during training, the updates on model weights and masks form a bi-level structure: ① depending on the magnitude of the updated weights, the topology of the mask is then updated, which forms the upper-level problem, ② depending on the newly obtained masks, the weights of the subnetwork will be updated through the standard SGD process (Amari, 1993), which forms the lower-level problem. Both problems ① and ② are mutually

dependent on the outcome of one another, which indicates that the formation of the DST essentially falls within the bi-level optimization (Colson et al., 2005; Zhang et al., 2023) realm, and both should be solved analytically. However, current DST optimizes the weight objective with conventional SGD optimization, while leaving the mask objective updated using non-comprehensive heuristics (e.g., weight magnitude (Mocanu et al., 2018; Mostafa & Wang, 2019; Yuan et al., 2021; Liu et al., 2021; Yuan et al., 2022), gradient (Evci et al., 2020; Jayakumar et al., 2020; Schwarz et al., 2021), momentum (Dettmers & Zettlemoyer, 2019), etc.). Consequently, the search efficiency for the mask is relatively low, making it prone to sub-optimal training accuracy. For instance, the mask tends to *stop evolving early* due to the convergence of weights (Yuan et al., 2021). Increasing mask update frequency along with using large learning rate and batch size (Evci et al., 2020) may partially alleviate the issue but incurs high system overhead (i.e., the amount of time or resources required in addition to those for computation in an end-to-end training process) (Chen et al., 2023a). Additionally, heuristic mask updating loses the opportunity for exploring the topological space of a neural network where the *learned structure* contains latent expressiveness power (Ramanujan et al., 2020). Although recent works have observed the performance gain of subnetworks through optimization (Tai et al., 2022), studies of their adoption in the DST methods are still lacking.

In this paper, we scrutinize the optimization opportunities in every phase of the DST, and we argue that the training scheme of DST essentially forms a bi-level structure, in which both levels should be solved simultaneously through optimization techniques. As such, we present a novel framework **Bi**-level **D**ynamic **S**parse **Tr**aining (**BiDST**), which for the first time, matures the partial-heuristic DST scheme into a fully optimization-based methodology. In BiDST, as shown in Figure 1 (b), the mask is optimized based on both weight and its topology, which are intercorrelated via implicit gradient (Krantz & Parks, 2002). We show that our approach not only provides a theoretically-grounded solution to-

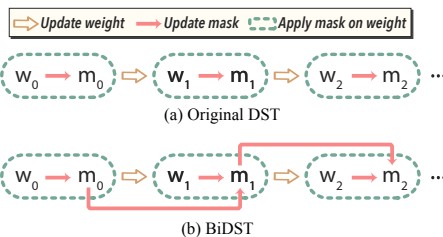

Figure 1: Training and mask update comparison between (a) original DST methods, and (b) the proposed BiDST.

wards sparse training, but also incurs only negligible extra computations that can be easily compensated through adjusting the training schedule without degrading accuracy. We demonstrate that BiDST shows superior mask searching quality and efficiency. Compared to prior DST methods, BiDST achieves notably higher accuracy while obtaining a more meaningful mask that can be used to identify a subnetwork with better training dynamics. More importantly, the optimization-based methodology serves as the driving force behind a more efficient mask searching procedure, which alleviates the system overhead problem that is primarily caused by frequently changing mask topology in DST, thus providing a more practical way to achieve efficient training. To summarize our contributions:

- We scrutinize every phase in the DST methods and find the lost treasure – an optimization opportunity on the mask update part that is long-neglected. This opportunity should be incorporated to reform DST into a bi-level structure, enabling comprehensive optimization.

- We design a novel sparse training framework – BiDST, which adopts bi-level optimization techniques to solve the weight and mask objective functions simultaneously while only incurring negligible computation costs.

- We demonstrate that BiDST excels in efficiently exploring the mask searching space, thus discovering superior sparse patterns that result in higher accuracy compared to other dynamic sparse training methods.

- We analyze that the frequent mask update is the source of the primary system overhead in the DST family, and demonstrate that the proposed BiDST framework is highly practical for the end-to-end implementation thanks to its optimization nature which reduces update frequency.

## 2 PROPOSED METHOD

### 2.1 NOTATIONS AND PRELIMINARIES

Consider a network function $f(\cdot)$ that is initialized as $f(x; \boldsymbol{\theta}_0)$ where $x$ denotes input training samples. We use $\boldsymbol{\psi}$ to define a sparse mask $\boldsymbol{\psi} \in \{0, 1\}^{|\boldsymbol{\theta}|}$ that is obtained from certain pruning algorithm, and

a sparse subnetwork is defined as $(\boldsymbol{\psi} \odot \boldsymbol{\theta})$ where $\odot$ is the element-wise multiplication. We use $s$ to denote the sparsity ratio, which is defined as the percentage of the *pruned* weights in the DNN model. For an entire training process, $t \in (1, T)$ denotes a certain iteration within the total $T$ iterations. We use $\ell$ to characterize the loss of the function $f(\cdot)$. By minimizing $\ell(f(\boldsymbol{\psi} \odot \boldsymbol{\theta}))$, a learned sparse subnetwork can be written as $f(\boldsymbol{\psi}_T \odot \boldsymbol{\theta}_T)$, where $x$ is omitted for the ease of notation purpose.

**Dynamic sparse training (DST).** To reduce the training computation cost and save hardware resources, DST trains a randomly initialized sparse neural network $f(\boldsymbol{\psi}_0 \odot \boldsymbol{\theta}_0)$ from scratch, and adjusts the sparsity topology $\boldsymbol{\psi}$ during training. The training scheme of a classic DST can be summarized as a process of alternating the following two phases:

- *Weight training:* based on the developed mask $\boldsymbol{\psi}_{t-1}$, we train the network for $t$ iterations, arriving at weights $\theta_t$ and network function $f(\boldsymbol{\psi}_{t-1} \odot \boldsymbol{\theta}_t)$.
- *Mask updating*: based on the trained weights $\boldsymbol{\theta}_t$, we use a mask updating function `MaskUpdate` to change the network topology to a better structure, i.e., $\boldsymbol{\psi}_t \leftarrow \texttt{MaskUpdate}(\boldsymbol{\psi}_{t-1}, \boldsymbol{\theta}_t)$.

In recent literature (Mostafa & Wang, 2019; Evci et al., 2020; Yuan et al., 2021), `MaskUpdate` is usually achieved by a two-step prune-grow process. First, the heuristic magnitude-based pruning prunes the weights to $(s + p)$, where $p$ is a hyper-parameter that increases the pruning ratio to a higher level. Then, a new mask $\boldsymbol{\psi}$ is obtained by growing a number of $p \times |\boldsymbol{\theta}|$ zero weights back, resetting the sparsity level back to $s$. Formally, the `MaskUpdate` can be defined as

$$f(\boldsymbol{\psi}_t \odot \boldsymbol{\theta}_t)|\boldsymbol{\psi}_t \leftarrow \texttt{ArgGrowTo}(\texttt{ArgPruneTo}(f(\boldsymbol{\psi}_{t-1} \odot \boldsymbol{\theta}_t), s + p), s) \tag{1}$$

**Bi-level optimization (BO).** Bi-level optimization (BO) serves as a valuable approach for modeling scenarios characterized by a two-tier hierarchy, wherein the upper-level variables exhibit a dependency on the outcomes derived from specific lower-level optimization problems. In general, a standard representation of BO can be expressed as

$$\underset{\boldsymbol{x} \in \mathcal{X}}{\text{minimize }} U(\boldsymbol{x}, \boldsymbol{v}^*(\boldsymbol{x})); \qquad \text{s.t. } \boldsymbol{v}^*(\boldsymbol{x}) \in \underset{\boldsymbol{v} \in \mathcal{V}}{\arg \min} \; L(\boldsymbol{x}, \boldsymbol{v}), \tag{2}$$

where $\mathcal{X}$ and $\mathcal{Y}$ are the feasible sets for the upper-level variable $\boldsymbol{x}$ and the lower-level variable $\boldsymbol{y}$, respectively. Consequently, this two-tier hierarchy can be characterized by two objectives as

- *Upper-level objective:* $U(\cdot)$ represents the upper-level objective, which aims to find an optimal solution $\boldsymbol{x}$ by considering the impact of the lower-level problem.
- *Lower-level objective:* $L(\cdot)$ represents the lower-level objective, which involves finding an optimal $\boldsymbol{y}$ that minimizes $L(\boldsymbol{x}, \boldsymbol{y})$ given $\boldsymbol{x}$.

The two levels of optimization are interdependent, and the solution to the upper-level problem depends on the outcome of the lower-level problem.

**Why dynamic sparse training is a bi-level optimization problem?** From the above discussion, it is clear that the two phrases in the DST training scheme are interdependent, that the development of mask is based on the trained weights while the topology of the weights is characterized by the developed mask. This process is very much similar to the formulation of a BO problem in equation 2. Considering the goal of DST is to train the weights while obtaining a good sparse topology (i.e., mask), the previous DST scheme can be informally written as

$$\texttt{MaskUpdate}(f(\boldsymbol{\psi}_{t-1} \odot \boldsymbol{\theta}_t)); \quad \text{s.t. } \boldsymbol{\theta}_t \in \arg \min \; \ell(f(\boldsymbol{\psi}_{t-1} \odot \boldsymbol{\theta}_{t-1})), \forall \, t \in (0, T), \tag{3}$$

where the dependency between $\boldsymbol{\psi}$ and $\boldsymbol{\theta}$ is characterized by how the solutions are obtained given certain conditions, i.e., given $\boldsymbol{\theta}_t$, `MaskUpdate`$(\cdot)$ finds $\boldsymbol{\psi}_t$, while $\boldsymbol{\theta}_t$ is trained from $\boldsymbol{\theta}_{t-1}$ that is structured by $\boldsymbol{\psi}_{t-1}$. It can be inferred that upper-level objective in equation 2 can be extended to $U(\boldsymbol{x}, \boldsymbol{v}^*(\boldsymbol{x})) \leftarrow \texttt{MaskUpdate}(f(\boldsymbol{\psi}_{t-1} \odot \boldsymbol{\theta}_t))$, and the lower-level objective in equation 2 can be extended to $L(\boldsymbol{x}, \boldsymbol{v}) \leftarrow \ell(f(\boldsymbol{\psi}_{t-1} \odot \boldsymbol{\theta}_{t-1}))$. Therefore, equation 2 is symbolically equivalent to equation 3, thus DST is essentially a bi-level optimization problem.

## 2.2 BiDST - Bi-Level Dynamic Sparse Training Framework

The intuition in Section 2.1 asserts that the DST is a BO problem, particularly because of the strong resemblance to the structure and characteristics of the latter one. In equation 3, we can see that the

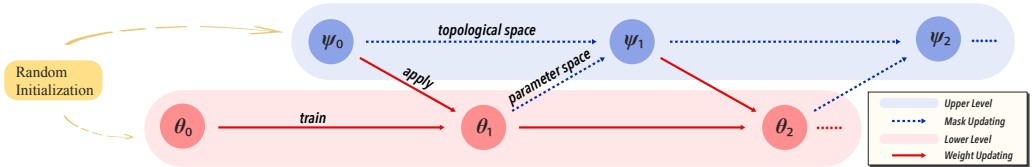

Figure 2: Overview of the two-level dynamic mask updating scheme of BiDST. The upper level learns the mask using topological space information from the previous mask and parameter space information from weights, and the lower level trains the sparse weights guided by mask.

current mask $\boldsymbol{\psi}_t$ is developed from $\boldsymbol{\psi}_{t-1}$ using current weights $\boldsymbol{\theta}_t$, and in the meantime, $\boldsymbol{\theta}_t$ is trained from $\boldsymbol{\theta}_{t-1}$ with previous topology guided by $\boldsymbol{\psi}_{t-1}$. The two phrases are interdependent on each other, where the mask determines which portions of the weights are subjected to training, and the weights, in turn, influence the subsequent sparse topology. Concretely, the solution to the upper-level problem in equation 3 is given by equation 1, which usually relies on rule-based heuristics that lead to sub-optimal solutions. Inspired by the fact that mask is embedded in the network computation graph due to its association with weights, we find that mask is implicitly characterized by a single unified loss function, and can be optimized through gradient-based techniques. Therefore, we rewrite equation 3 into the following bi-level dynamic sparse training (BiDST) formulation:

$$
\begin{aligned}
\underset{\boldsymbol{\psi} \in \boldsymbol{\Psi}}{\text{minimize}} \quad & \underbrace{\ell(f(\boldsymbol{\psi} \odot \overbrace{\boldsymbol{\theta}^*(\boldsymbol{\psi})}^{\text{Trained } \boldsymbol{\theta}}));}_{\text{Loss of the mask}} \\
\text{s.t.} \quad & \boldsymbol{\theta}^*(\boldsymbol{\psi}) \in \underbrace{\underset{\boldsymbol{\theta} \in \mathbb{R}^{|\boldsymbol{\theta}|}}{\arg\min} \; \ell(f(\boldsymbol{\psi} \odot \boldsymbol{\theta})) + \frac{\lambda}{2} \left\| \boldsymbol{\psi} \odot \boldsymbol{\theta} \right\|^2,}_{\text{Optimize weight } \boldsymbol{\theta} \text{ given fixed mask } \boldsymbol{\psi}}
\end{aligned}
\tag{4}
$$

where $\ell$ is the cross-entropy loss, and the regularization term in the lower-level problem with hyperparameter $\lambda$ stabilizes the gradient flow (Shaham et al., 2015; Hong et al., 2020). Depending on different objective variables, equation 4 decomposes a DST problem into two sub-problems:

- *Upper-level problem:* given trained weights $\boldsymbol{\theta}$, the upper-level problem aims to find a better mask $\boldsymbol{\psi}$ by minimizing the loss in terms of the mask.

- *Lower-level problem:* given a fixed mask $\boldsymbol{\psi}$, the lower-level problem updates the subnetwork weights by minimizing the loss in terms of weights, which is usually achieved by SGD.

By changing the upper-level objective from `MaskUpdate(·)` to the loss function of the network, BiDST exhibits better mask search ability by systematically exploring the solution space. Comparing to the trial-and-error strategies used in prior DST methods, BiDST can find globally near-optimal solutions more efficiently. In BiDST, a single loss function couples the topological space (i.e., mask) and the parameter space (i.e., weights) for joint optimization (see Figure 1(b) and Figure 2), which leverages the latent expressiveness power of the *learned structure* of the neural network. Additionally, the proposed optimization approach only incurs negligible computations by reusing gradients in the computation graph, and can be practically implemented by using Gumbel-Softmax (Jang et al., 2017) (please see Section 2.3).

To solve the BiDST optimization problem, we apply chain rule on the mask loss function $\ell(f(\boldsymbol{\psi} \odot \boldsymbol{\theta}^*(\boldsymbol{\psi})))$, we have:

$$
\frac{d\ell(f(\boldsymbol{\psi} \odot \boldsymbol{\theta}^*(\boldsymbol{\psi})))}{d\boldsymbol{\psi}} = \nabla_{\boldsymbol{\psi}} \ell(f(\boldsymbol{\psi} \odot \boldsymbol{\theta}^*(\boldsymbol{\psi}))) + \frac{d\boldsymbol{\theta}^*(\boldsymbol{\psi})^\top}{d\boldsymbol{\psi}} \nabla_{\boldsymbol{\theta}} \ell(f(\boldsymbol{\psi} \odot \boldsymbol{\theta}^*(\boldsymbol{\psi}))),
\tag{5}
$$

where $\frac{d\ell(f(\boldsymbol{\psi} \odot \boldsymbol{\theta}^*(\boldsymbol{\psi})))}{d\boldsymbol{\psi}}$ denotes the full derivative of $\ell$, $\nabla_{\boldsymbol{\psi}} \ell(f(\boldsymbol{\psi} \odot \boldsymbol{\theta}^*(\boldsymbol{\psi})))$ and $\nabla_{\boldsymbol{\theta}} \ell(f(\boldsymbol{\psi} \odot \boldsymbol{\theta}^*(\boldsymbol{\psi})))$ denote the partial derivative of the loss function with respect to mask and weights, respectively. Therefore, the optimization of the upper-level problem jointly considers both the topological space ($\nabla_{\boldsymbol{\psi}}$) and the parameter space ($\nabla_{\boldsymbol{\theta}}$).

---

**Algorithm 1:** BiDST implementation details

---

**Input:** A DNN model with randomly initialized weight $\boldsymbol{\theta}_0$; a random mask $\boldsymbol{\psi}_0$ with sparsity $s$. A flag parameter $\mathcal{F}$ indicating when to change the subnetwork topology based on $\boldsymbol{\psi}$.
**Output:** A sparse model satisfying the target sparsity $s$.
Set $t = 0$. Set the number of non-zero weights to be $k = s \times |\boldsymbol{\theta}|$.
**while** $t < T$ **do**
    **if** $\mathcal{F}$ **then**
        └ Binary masking $\boldsymbol{\psi}'_t \leftarrow \texttt{Binarize}(\boldsymbol{\psi}_t, \texttt{argmax}(\boldsymbol{\psi}_t, k))$
    Train the subnetwork $f(\boldsymbol{\psi}'_t \odot \boldsymbol{\theta}_t)$ by solving Eq. 10.
    Update mask $\boldsymbol{\psi}$ by solving Eq. 11.
    $t = t + 1$.

---

According to the implicit function theory (Gould et al., 2016), we refer to $\frac{d\boldsymbol{\theta}^*(\boldsymbol{\psi})^\top}{d\boldsymbol{\psi}}$ as implicit gradient (IG). For ease of notations, the transpose operation $\top$ is omitted in the rest of the paper. In equation 5, with the first-order stationary condition of BO (Ghadimi & Wang, 2018), we have

$$\nabla_{\boldsymbol{\psi}}\ell(f(\boldsymbol{\psi} \odot \boldsymbol{\theta}^*(\boldsymbol{\psi}))) + \lambda\boldsymbol{\theta}^* = 0. \tag{6}$$

By taking derivative with respect to $\boldsymbol{\psi}$ on both sides, we have:

$$\nabla^2_{\boldsymbol{\psi}\boldsymbol{\theta}}\ell(f(\boldsymbol{\psi} \odot \boldsymbol{\theta}^*(\boldsymbol{\psi}))) + \frac{d\boldsymbol{\theta}^*(\boldsymbol{\psi})}{d\boldsymbol{\psi}}\nabla^2_{\boldsymbol{\theta}}\ell(f(\boldsymbol{\psi} \odot \boldsymbol{\theta}^*(\boldsymbol{\psi}))) + \lambda\frac{d\boldsymbol{\theta}^*(\boldsymbol{\psi})}{d\boldsymbol{\psi}} = 0. \tag{7}$$

So that we can get

$$\frac{d\boldsymbol{\theta}^*(\boldsymbol{\psi})}{d\boldsymbol{\psi}} = -\nabla^2_{\boldsymbol{\psi}\boldsymbol{\theta}}\ell(f(\boldsymbol{\psi} \odot \boldsymbol{\theta}^*(\boldsymbol{\psi})))(\nabla^2_{\boldsymbol{\theta}}\ell(f(\boldsymbol{\psi} \odot \boldsymbol{\theta}^*(\boldsymbol{\psi}))) + \lambda\mathbf{I})^{-1}. \tag{8}$$

Since directly computing the second-order partial derivatives is theoretically challenging, inspired by Finn et al. (2017) and Liu et al. (2022a), we replace $\nabla^2_{\boldsymbol{\theta}}$ with zero and therefore

$$\frac{d\ell(f(\boldsymbol{\psi} \odot \boldsymbol{\theta}^*(\boldsymbol{\psi})))}{d\boldsymbol{\psi}} = \nabla_{\boldsymbol{\psi}}\ell(f(\boldsymbol{\psi} \odot \boldsymbol{\theta}^*(\boldsymbol{\psi}))) - \frac{1}{\lambda}\nabla_{\boldsymbol{\psi}\odot\boldsymbol{\theta}^*(\boldsymbol{\psi})}\ell(f(\boldsymbol{\psi} \odot \boldsymbol{\theta}^*(\boldsymbol{\psi}))). \tag{9}$$

At this point, solving the upper-level problem for mask update is to compute the first-order partial derivatives. Based on Hong et al. (2023), the analytical solution for the BiDST defined in equation 4 is derived following the classic DST regime in Section 2.1 as

- *Weight training:* at iteration $t$, we solve the *lower-level* problem using SGD, which is given by

$$\boldsymbol{\theta}_t = \boldsymbol{\theta}_{t-1} - \alpha\nabla_{\boldsymbol{\theta}}[\ell(f(\boldsymbol{\psi} \odot \boldsymbol{\theta})) + \frac{\lambda}{2}\|\boldsymbol{\psi} \odot \boldsymbol{\theta}\|^2]_{t-1}. \tag{10}$$

- *Mask updating*: after obtaining $\boldsymbol{\theta}_t$, we solve the *upper-level* problem with

$$\boldsymbol{\psi}_t = \boldsymbol{\psi}_{t-1} - \alpha\frac{d\ell(f(\boldsymbol{\psi} \odot \boldsymbol{\theta}_t))}{d\boldsymbol{\psi}}\big|_{\boldsymbol{\psi}=\boldsymbol{\psi}_{t-1}} \tag{11}$$

where $\alpha$ is the learning rate. We set $\alpha$ for upper- and lower-level solution to be the same for easy implementation. Algorithm 1 briefly shows the training process of the proposed BiDST.

## 2.3 COMPUTATION ANALYSIS FOR BiDST

In BiDST, the key parameters within the computation graph (e.g., weight, mask) are performing sparse computation (i.e., in both forward and backward propagation) to reduce the training cost. From Algorithm 1, weight and mask are two components that majorly contribute to the computation budget. Compared to prior DST methods, BiDST relaxes the mask variable to a continuous variable and computes its gradients, thus involving extra computation that arises from optimizing the mask. To minimize such extra cost, we apply `argmax(·)` function on mask to index a subset of weights for forward propagation. During backward propagation, mask gradients are derived from the same copy of activation gradients that is used to compute weight gradients. Therefore, mask doesn't take

Table 1: Accuracy comparison using ResNet-32 on CIFAR-10/100.

| Datasets | Sparsity Distribution | CIFAR-10 (dense: 94.9) | | | CIFAR-100 (dense: 74.9) | | |
|---|---|---|---|---|---|---|---|
| Pruning ratio | | 90% | 95% | 98% | 90% | 95% | 98% |
| LT [12] | non-uniform | 92.31 | 91.06 | 88.78 | 68.99 | 65.02 | 57.37 |
| SNIP [24] | non-uniform | 92.59 | 91.01 | 87.51 | 68.89 | 65.02 | 57.37 |
| GraSP [46] | non-uniform | 92.38 | 91.39 | 88.81 | 69.24 | 66.50 | 58.43 |
| Deep-R [2] | non-uniform | 91.62 | 89.84 | 86.45 | 66.78 | 63.90 | 58.47 |
| SET [31] | non-uniform | 92.30 | 90.76 | 88.29 | 69.66 | 67.41 | 62.25 |
| DSR [34] | non-uniform | 92.97 | 91.61 | 88.46 | 69.63 | 68.20 | 61.24 |
| RigL 10 | ERK | 93.55 | 92.39 | 90.22 | 70.62 | 68.47 | 64.14 |
| RigL-ITOP 28 | ERK | 93.70 | 92.78 | 90.40 | 71.16 | 69.38 | 66.35 |
| **BiDST (ours)** | ERK | **94.12±0.11** | **93.26±0.12** | **92.21±0.09** | **73.42±0.14** | **71.50±0.07** | **68.19 ±0.15** |
| RigL-ITOP [28] | uniform | 93.19 | 92.08 | 89.36 | 70.46 | 68.39 | 64.16 |
| RigL [10] | uniform | 93.07 | 91.83 | 89.00 | 70.34 | 68.22 | 64.07 |
| MEST [47] | uniform | 92.56 | 91.15 | 89.22 | 70.44 | 68.43 | 64.59 |
| **BiDST (ours)** | uniform | **93.68±0.11** | **92.93±0.14** | **91.99±0.12** | **72.97±0.08** | **70.03±0.12** | **67.21 ±0.06** |

any input for forward computation, and we certainly do not need to re-compute activation gradients when applying chain rule. We use the computational-modest Gumbel-Softmax (Jang et al., 2017) to ensure that the gradients flow smoothly for mask learning. Specifically, BiDST only learns a subset of mask that covers the current DNN topology and an extra space for mask updating (i.e., please see Appendix A for settings and discussion), which not only guarantees sparse computation on regular DNN computation, but also ensures sufficient search space (Lee et al., 2014) for mask updating.

## 2.4 OVERHEAD REDUCTION FOR PRACTICAL IMPLEMENTATION

Unlike other approaches that frequently change mask topology in their DST schedule (Evci et al., 2020; Mocanu et al., 2018), BiDST learns the subset of relaxed mask continuously but only changes the network topology periodically (i.e., `Binarize` through `argmax` operation of mask, please refer to $\mathcal{F}$ in Algorithm 1 and its setting in Appendix A). In practice, frequently changing the network topology incurs huge system overhead, which means more time and hardware resources are required to perform end-to-end learning due to computation graph reconstruction and static machine code recompilation (Chen et al., 2018). We show that BiDST only requires limited topology updates thanks to more principled and data-driven solutions (please see results in Section 3.4), which significantly reduces overhead and promotes practical implementations.

## 3 EXPERIMENTAL RESULTS

In this section, we carry out experiments to comprehensively demonstrate the advantages of BiDST. We evaluate BiDST in comparison with the state-of-the-art (SOTA) DST methods, and show superior accuracy, effective mask searching ability, as well as great applicability for implementations. We follow the traditional network and dataset selection used in prior DST methods. We use ResNet-32 (Zagoruyko & Komodakis, 2016), VGG-19 (Simonyan & Zisserman, 2014) and MobileNet-v2 (Sandler et al., 2018) on CIFAR-10 and CIFAR-100 datasets (Krizhevsky, 2009), and we use ResNet-34 and ResNet-50 (He et al., 2016) on ImageNet-1K dataset (Deng et al., 2009). We test the on-device training performance using a Samsung Galaxy S21 with Snapdragon 888 chipset. We repeat training experiments for 3 times for all experiments.

## 3.1 EXPERIMENTAL SETTINGS

We use standard training recipe following Yuan et al. (2021) and Wang et al. (2020b). To ensure fair comparison, all BiDST experiments have a slight scale down on the number of training epochs to compensate the mask learning computation cost. We use standard data augmentation, and cosine annealing learning rate schedule is used with SGD optimizer. For CIFAR-10/100, we use a batch size of 64 and set the initial learning rate to 0.1. For ImageNet-1K, we use a batch size of 1024 and learning rate of 1.024 with a linear warp-up for 5 epochs. Due to limited space, we put detailed settings in Appendix A.

Table 2: Accuracy comparison using ResNet-50 on ImageNet-1K with uniform sparsity.

| Method | Top-1 Accuracy (%) | Training FLOPS (×e18) | Inference FLOPS (×e9) | # of Mask Updates | Top-1 Accuracy (%) | Training FLOPS (×e18) | Inference FLOPS (×e9) | # of Mask Updates |
|---|---|---|---|---|---|---|---|---|
| Dense | 77.1 | 4.8 | 8.2 | n/a | - | - | - | - |
| Sparsity ratio | | 80% | | | | 90% | | |
| SNIP [24] | 69.7 | 1.67 | 2.8 | Static | 62.0 | 0.91 | 1.9 | Static |
| GraSP [45] | 72.1 | 1.67 | 2.8 | Static | 68.1 | 0.91 | 1.9 | Static |
| DeepR [2] | 71.7 | n/a | n/a | n/a | 70.2 | n/a | n/a | n/a |
| SNFS [9] | 73.8 | n/a | n/a | n/a | 72.3 | n/a | n/a | n/a |
| DSR [34] | 73.3 | 1.28 | 3.3 | n/a | 71.6 | 0.96 | 2.5 | n/a |
| SET [31] | 72.6 | 0.74 | 1.7 | n/a | 70.4 | 0.32 | 0.9 | n/a |
| RigL [10] | 74.6 | 0.74 | 1.7 | 320 | 72.0 | 0.39 | 0.9 | 320 |
| MEST$_{0.67\times}$ [47] | 75.4 | 0.74 | 1.7 | 45 | 72.6 | 0.39 | 0.9 | 45 |
| SpFDE [48] | 75.4 | 0.74 | 1.7 | 45 | - | - | - | - |
| **BiDST$_{0.67\times}$ (ours)** | **75.87** | 0.74 | 1.7 | 12 | **72.98** | 0.39 | 0.9 | 12 |
| MEST [47] | 75.7 | 1.27 | 1.7 | 60 | 75.0 | 0.65 | 0.9 | 60 |
| RigL$_{2\times}$* | 75.5 | 1.49 | 1.7 | 640 | 74.6 | 0.78 | 0.9 | 640 |
| Top-KAST [21] | - | - | - | - | 73.0 | 0.63 | 0.9 | 320 |
| **BiDST (ours)** | **76.15** | 1.26 | 1.7 | 18 | **75.39** | 0.65 | 0.9 | 18 |
| MEST$_{1.7\times}$ | 76.7 | 1.84 | 1.7 | 110 | 75.9 | 0.80 | 0.9 | 110 |
| RigL$_{5\times}$ [10] | 76.6 | 3.65 | 1.7 | 1,600 | 75.7 | 1.95 | 0.9 | 1,600 |
| **BiDST$_{1.7\times}$ (ours)** | **76.98** | 1.84 | 1.7 | 31 | **76.26** | 0.88 | 0.9 | 31 |
| MEST$_{8\times}$* | 77.6 | 8.80 | 1.7 | 550 | 77.0 | 3.84 | 0.9 | 550 |
| RigL$_{12\times}$* | 77.4 | 8.90 | 1.7 | 3,840 | 76.8 | 4.69 | 0.9 | 3,840 |
| C-GaP [30] | 77.9 | 12.07 | 1.7 | 28 | 76.3 | 9.69 | 0.9 | 28 |
| **BiDST$_{8\times}$ (ours)** | **78.01** | 8.80 | 1.7 | 146 | **77.38** | 3.85 | 0.9 | 146 |

* Our implementation with original source code using long training time.

## 3.2 BiDST ACCURACY

**CIFAR-10 and CIFAR-100.** We test BiDST using uniform and ERK (Evci et al., 2020) sparsity with 90%, 95%, and 98% overall sparsity. We demonstrate the accuracy with standard deviation results of ResNet-32 in Table 1. Due to space limitation, the results of all ERK sparsity, VGG-19 and MobileNet-v2 are included in Appendix B. According to the results, BiDST establishes a new state-of-the-art accuracy bar for various sparse training methods. When compared to state-of-the-art DST methods such as RigL and MEST, BiDST consistently outperforms the prior approaches by a significant margin. At varying levels of sparsity, we can see that BiDST achieves higher accuracy compared to the best baseline accuracy on CIFAR-10 by 0.49%, 1.1% and 2.62%, respectively, and on CIFAR-100 by 2.52%, 2.12% and 2.62%, respectively. With VGG-19 and MobileNet-v2, similar outperforming results can also be observed. Note that MobileNet-v2 is commonly considered a compact network that is hard to incorporate sparsity, BiDST still achieves promising results, which indicates that a more meaningful mask, though hard to find, can still be discovered by our method.

**ImageNet-1K.** Table 2 shows ResNet-50 on ImageNet-1K accuracy with uniform sparsity (ResNet-34 results in Appendix B). We stress all experiments are performed *three times*. We report the average accuracy and omit the standard deviation since the training is relatively stable (i.e., usually less than $0.1\%$ standard deviation) on large-scale datasets. Once again, BiDST achieves dominating performance against all baseline methods with regular training recipe. At similar training FLOPs, BiDST achieves significantly higher accuracy and requires fewer mask updates. When increasing the training time, BiDST exhibits promising accuracy gain. We extend training epochs to match RigL$_{2\times}$ (200 epochs), MEST$_{1.7\times}$ (250 epochs), RigL$_{5\times}$ (500 epochs), C-GaP (Ma et al., 2022) (990 epochs), MEST$_{8\times}$ and RigL$_{12\times}$ (1200 epochs). Note that C-GaP has partially-dense model during training, which has notably higher training FLOPs.

**Extreme Sparsity.** In dynamic sparse training, layer-wise sparsity ratios are pre-defined (e.g., uniform or ERK) to avoid layer collapse (Tanaka et al., 2020) in high sparsity (i.e., an entire layer is pruned). Nonetheless, different layers still experience different learning dynamics (Chen et al., 2023b), and high sparsity may prevent gradient from flowing smoothly. We conduct extreme sparsity ratios (e.g., up to $s = 99.9999\%$) on BiDST and compare the results to different DST methods. As shown in Figure 3, BiDST demonstrates better stability in extreme sparsity, indicating that a more

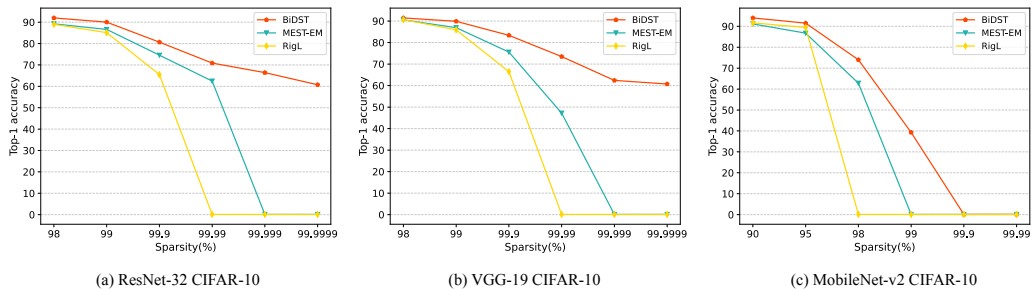

Figure 3: We perform high sparsity experiments on ResNet-32, VGG-19 and MobileNet-v2 on CIFAR-10 dataset, and compare with different DST methods. Note that MobileNet-v2 is already a compact model, so the extreme sparsity range is different from the other two networks.

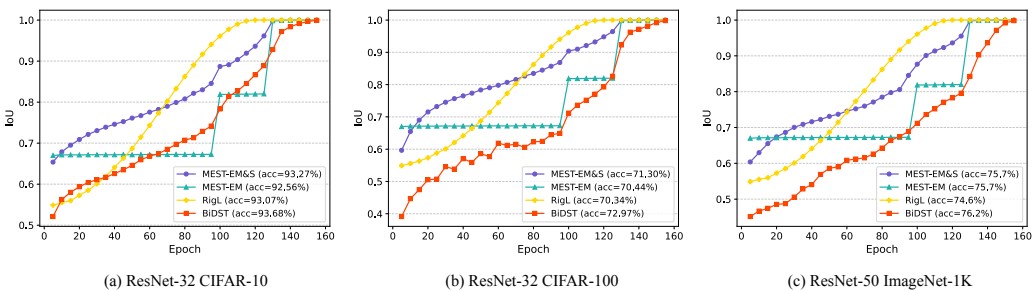

Figure 4: Evaluation of BiDST mask development progress. We use ResNet-32 on CIFAR-10/100, and ResNet-50 on ImageNet-1K at a target sparsity of 90%.

accurate mask can be identified from the network topology in any condition, such as large sparsity settings in training or when sparsifying a compact network structure (e.g., MobileNet family).

## 3.3 SPARSE MASK EVALUATION AND ANALYSIS

The optimization-based BiDST achieves higher mask searching efficiency compared to the prior heuristic-based approaches. For instance, the magnitude-based pruning and gradient (momentum) -based growing methods for mask updating may be significantly affected by the convergence of weights with the magnitude of the parameters oscillating within a very small range after certain times of training. Therefore, the commonly-used hard thresholding for such methods may not work effectively since the fluctuation is not sufficient enough to exceed certain thresholds. In BiDST, the mask updating is determined by jointly considering the importance of *both* network topology *and* weight value (i.e., please refer to Figure 2). We use Intersection over Union (IoU) to describe the development of mask during training process. As shown in Figure 4, at the same sparsity and within the same searching space, we compute the mask IoU of fixed intervals between two mask updates, and the results show that BiDST achieves better mask development (i.e., lower IoU means a higher portion of mask is updated). Compared to baseline methods that either suffer from high IoU or manually enforce the values within a certain range (e.g., MEST-EM&S), BiDST demonstrates a sufficient searching space and natural mask learning curves.

## 3.4 TRAINING SPEEDUP AND SYSTEM OVERHEAD OF DYNAMIC SPARSE TRAINING

To demonstrate practical implementation of BiDST, we extend the code generation of TVM (Chen et al., 2018) and design a training engine on the Snapdragon 888. For DST, the on-device computation is done by static code (e.g., OpenCL for GPU and C++ for CPU), and the training acceleration is obtained from compiler optimization that skips the zeros in weights. We set a target accuracy and perform experiments using representative DST methods. According to Figure 5, BiDST achieves the best training acceleration performance (i.e., highest sparsity) among all other methods. Compared to dense training, BiDST achieves 1.8-2.1× training speedup. Meanwhile, dynamic topology switching on the hardware requires code re-compilation, which is the source of the major system overhead in DST. The mask update frequency of BiDST is the lowest among all other methods, resulting

Figure 5: System overhead in DST when implemented on Snapdragon 888. We compare representative DST methods (e.g., uniform sparsity RigL (Evci et al., 2020), global reparameterization DSR (Mostafa & Wang, 2019)) with BiDST in uniform sparsity at similar accuracy.

in 16-25× reduced overhead time compared to other baselines. Additionally, the uniform sparsity of BiDST alleviates the resource allocation burden, which further contributes to better speedup compared to the global reparameterization method such as DSR (Mostafa & Wang, 2019).

# 4 RELATED WORKS

**Dynamic sparse training.** Parameter importance estimation plays a pivotal role in DST, aiding in the identification of parameters worthy of retention or pruning. Various heuristic-based methods are employed for this purpose. For example, SET (Mocanu et al., 2018) removes the least magnitude-valued weights during training and regrows an equivalent number of weights randomly at the end of each epoch. SNFS (Dettmers & Zettlemoyer, 2019) uses exponentially smoothed momentum to identify important weights and layers, redistributing pruned weights based on mean momentum magnitude. RigL (Evci et al., 2020) updates the sparsity topology of the network during training using magnitude-based weight dropping method and regrows weights using the top-k absolute largest gradients. ITOP (Liu et al., 2021) highlights the benefits of dynamic mask training, considering all possible parameters across time. Top-KAST (Jayakumar et al., 2020) offers a scalable and consistently sparse DST framework for improved effectiveness and efficiency. MEST (Yuan et al., 2021) employs a gradually decreasing drop and grow rate with a more relaxed range of parameters for growing. AD/AC (Peste et al., 2021) proposes a co-training of dense and sparse models method, which generates accurate sparse–dense model pairs at the end of the training process. ToST(Jaiswal et al., 2022) provide a plug and play toolkit to find a better sparse mask for current pruning method. Nowak et al. (2023) proposes that pruning criterion affects the update of network typology, thus impacting the final performance.

**Heuristic- and Optimization-based Sparsity Learning.** There are various heuristic-based methods for learning a sparse network, such as (Han et al., 2015; Molchanov et al., 2017; 2019; Frankle & Carbin, 2018). To improve the performance, data-driven approaches emerged to better optimize the mask finding process. SNIP (Lee et al., 2019) sparsifies the network at early stage of training by studying gradients of the training loss at initialization. GraSP (Wang et al., 2020b) preserves the gradient flow and prunes the network at initialization. SynFlow (Tanaka et al., 2020) studies the layer-collapse issue in sparse training and propose to use network synaptic flow to guide early pruning of the network. Additionally, numerous studies incorporate optimization techniques. ADMM (Zhang et al., 2018; Ren et al., 2019) transfers the non-convex optimization problem into two sub-problems that are solved iteratively. OLMP (Li et al., 2018) transforms the threshold tuning problem into a constrained optimization problem and employs powerful derivative-free optimization algorithms to solve it. Spartan (Tai et al., 2022) combines soft masking and dual averaging-based updates, enabling flexible sparsity allocation policies. Liu et al. (2022b) uses the energy estimate of each layer as the Frobenius norm optimization constraint for the convolution kernel weights.

# 5 CONCLUSION

In this paper, we propose a novel design to establish a full optimization-based dynamic sparse training framework BiDST, which reformulates the prior partial-heuristic-based dynamic sparse training into a bi-level optimization problem, and solves the problem of each level (e.g., mask finding and weight training) simultaneously and analytically. BiDST achieves prominent accuracy in the parameter-sparse training regime due to a more principled and data-driven approach, while promoting great applicability for on-device implementation by reducing system overhead brought by dynamism in DST methods.

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

# Appendix

## A EXPERIMENT SETTINGS

Table A.1 lists training hyerparatemer settings for BiDST. For on-device speed evaluation, we scale down batch size to 64 for all network and dataset. Since we can use gradient accumulation to obtain larger batch size training, the accuracy for on-device experiments is not affected by different batch size. To make BiDST search efficiently, we use an extra space for mask to learn, which constitutes a very small portion of the mask beyond the space occupied by existing non-zero weights. We define this small portion of search space as $\mathcal{S}$, which is the ratio of the number of extra mask values to the total number of mask elements. During training, this small portion of mask is updating normally, but their corresponding weights and weight gradients are zeros. Therefore, this part of the mask is learning the importance of structure. Please note that mask learning in BiDST *does not* need any input, and reuses the activation gradient with Gumbel-Softmax for backward propagation, thus incurring very limited computation compared to traditional convolution operation. The epoch settings in Table A.1 can reflect how efficient our mask learning scheme is: we scale down the training time to match the training computation with other baselines, and the degree of epoch reduction is small compared to the total training epochs. For instance, our training epoch setting inherits MEST (Yuan et al., 2021) and GraSP (Wang et al., 2020a) as 160 epochs to train CIFAR-10/100, but we only train BiDST 158 epochs to compensate those extra computation of mask learning. For ImageNet-1K, we scale down training time from 150 epochs to 148 epochs to match computation with our comparison baselines.

Table A.1: Hyperparameter settings.

| Experiments | VGG-19 | ResNet-32 | MobileNet-v2 | ResNet-50/34 |
|---|---|---|---|---|
| Dataset | CIFAR | CIFAR | CIFAR | ImageNet-1K |
| Training hyperparameter settings | | | | |
| Training epochs | 158 | 158 | 347 | 148 |
| Batch size | 64 | 64 | 64 | 1024 |
| Learning rate scheduler | cosine | cosine | cosine | cosine |
| Initial learning rate | 0.1 | 0.1 | 0.1 | 1.024 |
| Ending learning rate | 4e-8 | 4e-8 | 4e-8 | 0 |
| Momentum | 0.9 | 0.9 | 0.9 | 0.875 |
| $\ell_2$ regularization | 5e-4 | 1e-4 | 5e-5 | 3.05e-5 |
| Warmup epochs | 5 | 0 | 5 | 8 |
| BiDST hyperparameter settings | | | | |
| Mask update frequency | 8 | 8 | 8 | 5 |
| Mask search space ($\mathcal{S}$) | $s = 0.90, \mathcal{S} = 0.05$ 
 $s = 0.95, \mathcal{S} = 0.03$ 
 $s = 0.98, \mathcal{S} = 0.02$ | $s = 0.90, \mathcal{S} = 0.05$ 
 $s = 0.95, \mathcal{S} = 0.03$ 
 $s = 0.98, \mathcal{S} = 0.02$ | $s = 0.90, \mathcal{S} = 0.08$ 
 $s = 0.95, \mathcal{S} = 0.05$ 
 $s = 0.98, \mathcal{S} = 0.04$ | $s = 0.90, \mathcal{S} = 0.05$ 
 $s = 0.95, \mathcal{S} = 0.03$ 
 $s = 0.98, \mathcal{S} = 0.02$ |
| Regularization coefficient $\lambda$ | 1e-4 | 1e-4 | 1e-4 | 5e-5 |

## B EXPERIMENTAL RESULTS CONTINUE

### B.1 CIFAR-10 AND CIFAR-100 ON VGG-19 AND MOBILENET-V2

We inlude the extra experimental results in this section. We demonstrate the results of VGG-19 and MobileNet-v2 on CIFAR-10/100, and both with uniform and ERK sparsity scheme. Table B.1 shows the VGG-19 results and Table B.2 shows the MobileNet-v2 results. As MobileNet-v2 is a compact network structure, we follow the classic settings (Zhuang et al., 2018) of training MobileNet-v2 and set the training epoch to 350 for better loss convergence.

### B.2 IMAGENET-1K ON RESNET-34

Table B.3 shows the top-1 accuracy results of ResNet-34 on ImageNet-1K. For ImageNet-1K results, we use the uniform sparsity distribution. Similar to the settings in our main paper, we also extend

Table B.1: Test accuracy of VGG-19 on CIFAR-10/100.

| Methods | Sparsity Distribution | CIFAR-10 (dense: 94.2) | | | CIFAR-100 (dense: 74.2) | | |
|---|---|---|---|---|---|---|---|
| Pruning ratio | | 90% | 95% | 98% | 90% | 95% | 98% |
| LT [12] | non-uniform | 93.51 | 92.92 | 92.34 | 72.78 | 71.44 | 68.95 |
| SNIP [24] | non-uniform | 93.63 | 93.43 | 92.05 | 72.84 | 71.83 | 58.46 |
| GraSP [46] | non-uniform | 93.30 | 93.43 | 92.19 | 71.95 | 71.23 | 68.90 |
| Deep-R [2] | non-uniform | 90.81 | 89.59 | 86.77 | 66.83 | 63.46 | 59.58 |
| SET [31] | non-uniform | 92.46 | 91.73 | 89.18 | 72.36 | 69.81 | 65.94 |
| DSR [34] | non-uniform | 93.75 | 93.86 | 93.13 | 72.31 | 71.98 | 70.70 |
| RigL-ITOP [28] | uniform | 93.19 | 92.08 | 89.36 | 70.46 | 68.39 | 64.16 |
| RigL [10] | uniform | 93.12 | 92.43 | 90.65 | 71.14 | 69.02 | 64.87 |
| MEST+EM [47] | uniform | 93.07 | 92.59 | 90.55 | 71.23 | 69.08 | 64.92 |
| **BiDST (ours)** | uniform | **93.76±0.20** | **93.18±0.17** | **91.45±0.19** | **72.26±0.21** | **69.87±0.16** | **65.82 ±0.22** |
| RigL [10] | ERK | 93.77 | 92.75 | 90.87 | 71.34 | 69.21 | 65.02 |
| RigL-ITOP [28] | ERK | 93.81 | 92.81 | 90.53 | 71.46 | 69.58 | 66.72 |
| **BiDST (ours)** | ERK | **94.04±0.21** | **93.74±0.24** | **91.52±0.19** | **72.31±0.19** | **70.94±0.15** | **69.01 ±0.22** |

the training time to $5\times$ training epochs. We can see from the results that the accuracy increase is consistently observed for $5\times$ setting, but not similarly significant compared to Resnet-50 network. We believe that the reason is that ResNet-34 is smaller than ResNet-50, thus the capacity of the network is not sufficient enough for fully capturing the complex patterns and features in the data.

Table B.2: MobileNet-v2 on CIFAR-10/100.

| Method | Sparsity Distribution | Sparsity Ratio | Test Accuracy | Sparsity Ratio | Test Accuracy |
|---|---|---|---|---|---|
| | | CIFAR-10 | | CIFAR-100 | |
| *Dense accuracy:* | | 94.1% | | 73.5% | |
| BiDST | Uniform | 0.9 | 91.8 | 0.95 | 71.0 |
| | Uniform | 0.95 | 91.1 | 0.95 | 69.8 |
| | Uniform | 0.98 | 89.7 | 0.98 | 68.3 |
| BiDST | ERK | 0.9 | 92.3 | 0.9 | 71.7 |
| | ERK | 0.95 | 91.7 | 0.95 | 70.6 |
| | ERK | 0.98 | 90.4 | 0.98 | 69.8 |

Table B.3: ResNet-34 on ImageNet.

| Method | Sparsity Distribution | Sparsity Ratio | Test Accuracy |
|---|---|---|---|
| ImageNet-1K (*dense top-1 accuracy: 74.1%*) | | | |
| BiDST | Uniform | 0.6 | 74.66 |
| | Uniform | 0.7 | 74.35 |
| | Uniform | 0.8 | 73.99 |
| | Uniform | 0.9 | 73.08 |
| BiDST$_{5\times}$ | Uniform | 0.6 | 74.82 |
| | Uniform | 0.7 | 74.59 |
| | Uniform | 0.8 | 74.20 |
| | Uniform | 0.9 | 73.19 |

# C ABLATION STUDY

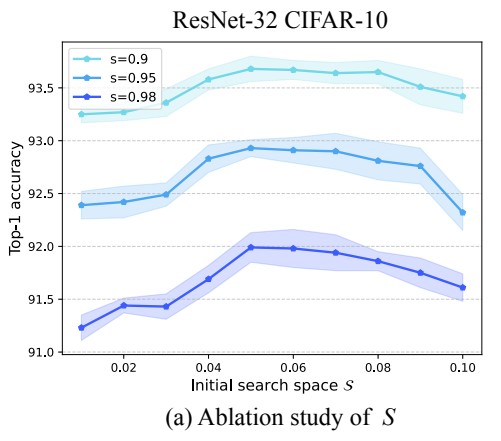

(a) Ablation study of $S$

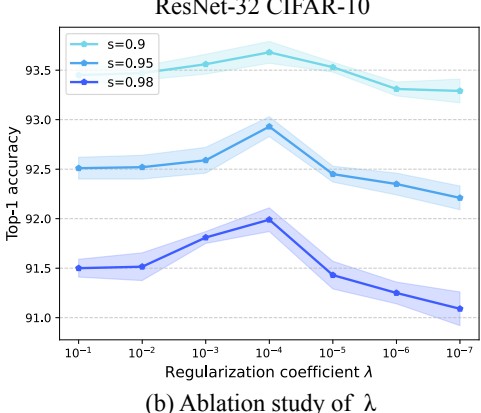

(b) Ablation study of $\lambda$

Figure C.1: Ablation study of the important hyperparameters of BiDST.

❖ **Mask search space** $\mathcal{S}$ **variation.** In BiDST, the mask search space is an extra space for mask to learn. The size of this space is related to the efficiency of the mask learning procedure. We manually set $\mathcal{S}$ with different value, and test the accuracy of ResNet-32 on CIFAR-10. We set the searching space ranging from 0.01 to 0.1, and show the results with different sparsity in Figure C.1 (a). As we can notice, the search space is not the larger the better. We choose to use the current setting because smaller space incurs less extra computation. We leave the research of BiDST search space optimization for future study.

❖ **Regularization coefficient** $\lambda$. We test BiDST at different regularization coefficient $\lambda$ in Figure C.1 (b). $\lambda$ controls the strength of a regularization term that discourages drastic changes in the mask by penalizing large gradients. A smaller $\lambda$ allows for larger changes in the mask, while a larger $\lambda$ enforces a stronger constraint on the changes to the mask. We set $\lambda$ from 1e-7 to 1e-1 and perform BiDST with ResNet-32 on CIFAR-10. Please note that a larger change in mask update is not necessarily beneficial for accuracy (e.g., MEST-EM enforce strict search space). The high accuracy of BiDST is due to the optimization that systematically connects parameter space and topological space.

