# OpenReview forum: "BiDST: Dynamic Sparse Training is a Bi-Level Optimization Problem"
_ICLR.cc/2024/Conference — Submitted to ICLR 2024_

### Official Review · Reviewer_9sSt · 2023-10-30

**Soundness:** 2 fair
**Presentation:** 2 fair
**Contribution:** 2 fair
**Rating:** 3
**Confidence:** 4

**Summary:**

The authors formulate the dynamic sparse training problem as a bi-level optimization problem.  Based on the bi-level formulation, the authors propose an algorithm that solves the mask and weights in the network simultaneously.  Experiments show that the proposed algorithm outperforms the other methods.

**Strengths:**

The authors formulate the DST problem as a bi-level optimization problem. To solve the computational issue, the authors discard the hessian term in the gradient estimation, which does not result in any performance drop according to the experimental results. Further, to deal with the discrete issue of the mask, the authors introduce a condition to update the discrete mask. The experimental results show that the proposed algorithm outperforms the other methods.

**Weaknesses:**

1. The proposed algorithm does not have any convergence guarantee. Different from other bi-level optimization algorithms, the algorithm is based on a rough estimation of the gradient (eliminating the inversion of the second-order derivative) and is facing the discrete issue of the mask ( the estimation of the gradient is not at the point of the continuous variable of the mask but a hard discretization of the soft mask).  Thus, it is important to have a theoretical guarantee to ensure the performance of the proposed algorithm.

2. The upper-level objective is non-natural to me. Since the upper-level problem is also a finite sampling objective, we need some regularization to get good results. Meanwhile, the nature regularization is the l2 norm (i.e., $||\theta^*(\psi)||$).  With this regularization, we can formulate the upper-level problem as the same as the lower-level problem. Thus, the problem can be formulated into a single-level problem, which is easier to solve than a bi-level problem.  Further, people use bi-level formulation in machine learning usually when upper-level and lower-level objective contains different dataset,

**Questions:**

See weakness.

---

> ### Author Response · Authors · 2023-11-20
>
> We sincerely appreciate your thoughtful comments about our work. Here are our response.
>
> Q1. Convergence Guarantee
>
> A1. We appreciate the reviewer's consideration of convergence guarantees. It's crucial to note that the nature of dynamic sparse training methods, including our proposed BiDST, often involves challenges that make it inherently difficult to provide formal convergence guarantees. Many successful literature has demonstrated that it is not necessary to include such convergence analysis for DST since the performance of DST is already proved to be SOTA in the sparse training research domain. We humbly disagree with the opinion that the lack of convergence guarantee is the major reason for rejection since the uniqueness of BiDST lies in its ability to address the dynamic nature of sparse training by leveraging a rough estimation of the gradient and handling discrete mask issues. On the contrary, we innovatively adopt bi-level optimization with improved efficiency to solve the DST problem, while we also reduce the computation cost in the solution that approximates the discrete mask into continuous variables. These aspects set BiDST apart from conventional approaches.
>
> It's essential to recognize that existing methods in the DST domain also lack strict convergence guarantees due to the intricacies of optimizing sparse structures. Our results, demonstrating superior performance, speak to the effectiveness of BiDST in the context of dynamic sparse training. We believe that the focus should be on the empirical validation and real-world applicability of the algorithm, where BiDST excels, rather than imposing traditional convergence expectations that may not align with the unique challenges of sparse training.
>
>
> Q2. Non-Natural Upper-level Objective
>
> A2. We appreciate the reviewer's consideration of the upper-level objective. The upper-level problem is the optimization of the mask parameter based on the trained weights. In our implementation, our code uses one single optimizer, which adopts the standard L2 regularization with the penalty parameter that can be found in Table A.1 in our appendix. However, we do not believe it is necessary to incorporate such regularization terms in the equations. While regularization, including the traditional L2 norm, is a common practice in training neural networks, its omission from both objective formulations is a deliberate choice to maintain clarity and emphasize the bi-level optimization framework. For the lower-level problem, however, since we are optimizing sparse weights, we add the second term as a sparse regularization to stabilize the training without incurring dense computation of all parameters. This regularization is controlled by a coefficient hyper-parameter that can also be found in Table A.1.
>
> We also humbly disagree with the opinion that if we add the L2 regularization to the upper-level problem, the whole problem becomes the single-level optimization problem. Due to the nature of sparse training, there are two objectives that need to be optimized regardless of the method used for such optimization. Those two objectives are weights and masks. Even if we add L2 regularization and use it for mask/weight optimization, the effectiveness cannot be comparable to our design since using L2 to find mask is essentially through the penalization of all mask parameters, but our approach is to analytically learn an optimal mask based on all information flow in training.
>
> It's also crucial to emphasize that even with the addition of a regularization term in the upper level, the overall optimization, by our intentional design, remains a bi-level optimization problem in nature. We have provided clear illustrations, as shown in Figure 1, demonstrating that while other dynamic sparse training (DST) methods may be inherently bi-level, they lack the optimization techniques that BiDST employs. Our focus is on optimizing both levels simultaneously through our novel bi-level approach, which distinguishes BiDST from other sparse training methods.
>
> Lastly, it is not always true that bi-level optimization is used with different datasets for its upper and lower problem. Please see the literature of [R1]. We acknowledge that some works might use different datasets, but bi-level optimization is a very general solution framework, and can be used in different settings when using proper design. Therefore, we can embrace the possibility that bi-level optimization is a very useful tool to solve real-world problems, and that’s indeed what this paper’s focus is.
>
> [R1] Zhang, Yihua, et al. "Revisiting and advancing fast adversarial training through the lens of bi-level optimization." International Conference on Machine Learning. PMLR, 2022.

---

> > ### Comment · Reviewer_9sSt · 2023-11-22
> >
> > For a bilevel optimization problem like
> > \\[
> > \begin{aligned}
> > &min_x f(x,y^*(x))\\\\
> > &s.t. y^*(x) \in \arg\min_y f(x,y)
> > \end{aligned}
> > \\]
> > can be written as $min_x min_y f(x,y)$ then it becomes a single level problem.
> >
> > Similar to the setting, where a regularization is added. Apparently, when we add a regularization to the upper-level problem, the coefficient should be the same as it is in the lower-level problem. Thus, the function form of the upper-level objective function and the lower-level objective problem will be the same. Taking the previous formulation,  the bi-level problem can be reformulated into a single-level problem. Then some simple algorithm (e.g. SGD) can be applied to solve the problem.

---

### Official Review · Reviewer_MEAC · 2023-11-01

**Soundness:** 3 good
**Presentation:** 2 fair
**Contribution:** 3 good
**Rating:** 5
**Confidence:** 4

**Summary:**

Dynamic Sparse Training (DST) methods are the state-of-the-art in sparse training and sparse mask finding methods. They typically work by learning both a mask and weights for a model jointly during training, however while the model weights are learned using back propagation, the mask itself is typically learned using a heuristic such as random (SET) or gradient norm (RiGL). The authors propose a DST method, BiDST, to instead jointly optimize the mask and weights using an approximation to the bi-level optimization framework. The authors present state-of-the-art results on BiDST compared to existing state-of-the-art DST methods using ResNets on CIFAR-10/100 and ImageNet.

**Strengths:**

* The paper is well written in the majority, with a clear motivation, background and experimental setup. The unfortunate exception to this writing is some of the key parts of the methodology most important to comparing it with existing DST methods, and understanding the speedups claimed.
* One of the weaknesses of the current DST methods is the heuristic nature of mask updates, anything that gets away from heuristics on this front is a welcome development and the proposed method is indeed a data-driven approach.
* The bi-level methodology is intuitively suited to the DST framework, as described by the authors, at least in the original form.
* The experimental methodology, in terms of models, datasets, training regimes, sparsity levels and compared baselines is excellent, and it is obvious that the authors are very familiar with the existing DST literature from their usage in this, in particular the comparisons across different multiples of training epochs.
* Although the background is short, and awkwardly positioned in the paper, it does cover much of the relevant literature in terms of citations, if perhaps not explanation. Much of this is also covered implicitly in the methodology/results in comparing to existing methods.
* For the most part, the results would appear to be state-of-the-art for a DST method overall, and are very motivating for the proposed method, at least assuming that BiDST is in fact comparable to existing DST methods - i.e. it is in fact sparse training (not dense), and is not significantly more expensive (although the latter at least appears to not be true). The authors discuss that they attempt to make the results comparable by training BiDST for less time, which is good, although at the end of the day it's not clear to me how fair a comparison this is.

**Weaknesses:**

* The reason directly optimizing the mask is typically not done is of course that a (hard) binary mask is non-differentiable. Using a soft mask in any form would necessarily be dense, not sparse, training. I think the authors do not focus on explaining what they do with respect to both of these things enough in the paper as it is currently written, as I'm still not confident I understand how this is addressed by BiDST. I believe the authors use the Gumbel-Softmax trick, and while that might explain how they learn a binary mask using gradients, that would mean that BiDST uses dense training as far as I can see. And yet the authors demonstrate sparse training acceleration in their results (and this itself is unclear). I have asked this in the question section in more detail, and a comprehensive answer by the authors is vital in my being able to understand this paper better as a reader and a reviewer. Understanding the fact of if BiDST is in fact doing dense weight/mask updates is key in understanding if it is fair to compare BiDST with existing DST sparse training methods, such as SET, as the main motivation of these methods is to reduce compute during training.
* In the explanation below Equation 8 in section 2.2, the authors casually mention that they simply replace the second-order partial derivatives with 0 in their derivation of the weight/mask-update rules for BiDST. Isn't it precisely these second-order partial derivatives that carry much of the information of the relationship between the mask and weight parameters? There is no discussion on the effect of this, or why this is a reasonable thing to do aside from the fact it is expensive to calculate. Note that the first-order partials are retained. After all the fanfare of BiDST doing optimization "properly" compared to existing DST methods in the motivation, this is quite a let-down, and I think the authors should note this earlier/be careful of giving the impression of over-claiming and be much more transparent that they are crudely approximating the bi-level optimization methodology.
* As explained in 3.1, BiDST experiments are presented using fewer training epochs to ensure a "fair" comparison due to the "mask learning computation cost", however it's not detailed what exactly is the fair number of epochs to compare and why, and I didn't see any numbers or details on what the "mask learning computation cost" overhead is w.r.t. existing DST and other sparse training methods.
* It is not clear how significant some of the ImageNet results are, being within 0.2 percentage points of the baselines in some cases. Because of this it really stands out as suspicious that although the authors stress that ImageNet experiments are all performed three times, we are not given the variance across these three runs to aid us in understanding the significance of the results, as for example done with CIFAR-10/100. The authors say "we omit these because it's less than 0.1% std dev (is this percentage or percentage points?). Why not just show them in the table? I would not ask for multiple imagenet runs normally, but if you've performed them and are stressing that you did, why hold back on showing the variance?
* In section 3.3, the authors use IoU to analyze the evolution of the mask during training and compare to existing DST methods, claiming "BiDST achieves better mask development" based on low IoU compared to other DST methods. However, while potentially interesting, the motivation for this analysis is nowhere near convincing enough to make such a bold claim.
* In section 3.4, when explaining the "sparse training" speedups shown, the authors explain "training acceleration is obtained from compiler optimization that skips the zeroes weights". If this is a compiler optimization, it's static analysis, i.e. a fixed mask. How is this possible for a DST method that is changing masks potentially every iteration, or is the "training engine" actually to speed up inference timings? This explanation is important as speeding up unstructured sparsity on real-world hardware is something that is not easy, and in fact is well worth a standalone publication if it was truly being solved so easily by the authors.
* The background is very short, and awkwardly positioned in the paper. It appears much of the relevant background is distributed throughout the method implicitly by citing baselines and methodology, but personally I'm a fan of the traditional consolidated background before a method.

**Questions:**

I have currently rated the paper borderline as although the results are impressive, and the method appealing, I don't believe that the explanation of the method is detailed or clear enough right now to understand if this is in fact comparable to other sparse training/DST methods. I would appreciate detailed feedback from the authors to address the main points for which I am currently confused on after reading their paper:

* Is BiDST sparse training or dense training? i.e. are dense gradients or dense weights used during training? Please detail how explicitly. I'm confused because learning the mask necessarily means that the mask cannot be binary (it's approximated by Gumbel-Softmax I believe), and yet you show speedups during training due to "sparse training". Indeed the mask update rule in Equation 11 appears to be dense, and it would appear this is run every iteration. There is some text on selecting a subset of the mask, but that doesn't go into anywhere near as much detail as you need to make this point clear and understandable.
* In the explanation below Equation 8 in section 2.2, the authors casually mention that they simply replace the second-order partial derivatives with 0 in their derivation of the weight/mask-update rules for BiDST. Why this is a reasonable thing to do aside from the fact it is expensive to calculate? What is the effect on the optimization/ability to learn the mask/weights jointly?
* As explained in 3.1, BiDST experiments are presented using fewer training epochs to ensure a "fair" comparison, however it's not detailed what exactly is the fair number of epochs to compare and why. How many steps does a "single" BiDST iteration take compared to e.g. SET? Give numbers/details on what the "mask learning computation cost" overhead is w.r.t. existing DST and other sparse training methods.
* What exactly is the variance of the ImageNet results, not clear if stddev is 0.1% is percentage or percentage points.
* In section 3.4, if this is a compiler optimization, i.e static analysis, then it must necessarily be on a fixed mask? How is this possible for a DST method that is changing masks potentially every iteration, or is the "training engine" actually to speed up inference timings? If it is not static analysis/a fixed mask, explain in detail how speedups are achieved on real-world hardware for unstructured sparsity.

---

> ### Author Response · Authors · 2023-11-20
>
> **This is the 1/2 part of the response.**
>
> We thank the reviewer for your thoughtful comments about our work.
>
> Below are our responses about the first three questions.
>
> Q1. Sparse computation for BiDST.
>
> A1. According to Section 2.3, we discuss that the BiDST’s key parameters within the computation graph (e.g., weight, mask) are performing sparse computation (i.e., in both forward and backward propagation) to reduce the training cost. The reason is that we only learn partial mask parameters that associate with the active non-zero weights, and the update of mask parameters is also very efficient since it reuses the activation gradients and takes no input data for forward computation. For weights in the neural network, the forward computation is sparse according to equation 4’s lower equation that all terms are masked with the sparse mask. For backward computation, we only update the non-zero weights so that the entire computation is sparse. We also want reviewer to refer to reviewer xhao’s strengths part that acknowledges our method can achieve overall efficiency during sparse training. For the results that demonstrate on-device acceleration, we adopt the compiler-level optimization [R1] that transforms the DNN sparse computation graph with static execution code, thus achieving reduced compact computation. We will add our implementation details in our revision.
>
> Q2. Handling Second-order Partial Derivatives in BiDST
>
> A2. We acknowledge the reviewer's concern. The decision to replace second-order partial derivatives with 0 is made for computational efficiency reasons. We follow the method used in citation Finn et al. (2017) and Liu et al. (2022a) to achieve this conversion. However, we understand the need for a more detailed explanation. We will revise the paper to include a thorough discussion of the implications and potential limitations of this approximation. Transparency regarding this decision will be emphasized to provide a more accurate portrayal of the optimization methodology used in BiDST.
>
> Q3. Fair Comparison and Mask Learning Computation Cost
>
> A3. We understand your concern. The reduced training epochs is shown in Table A.1 in appendix. For the mask update, the only computation related to the overall training cost is the Gumbel-softmax, and it becomes the mask mask learning computation cost overhead to the overall training. Since Gumbel-softmax is a very efficient implementation which only needs element-wise computation instead of convolution computation, the overall cost for mask updating is approximately 1% of the total training cost when we train CIFAR on ResNet-32 with 160 epochs. Therefore, we reduce the training epochs by 2 epochs to match the training cost with other baselines that train 160 epochs.
>
> [R1] Niu, Wei, et al. "Patdnn: Achieving real-time dnn execution on mobile devices with pattern-based weight pruning." ASPLOS. 2020.
>
> **Please continue to the next part of the response.**

---

> > ### Comment · Reviewer_MEAC · 2023-11-22
> >
> > I'd like to thank the authors for their rebuttal, and apologize for my late participation in the rebuttal period - this was due to exceptional circumstances.
> >
> > Q1: This is unfortunately still very unclear to me as written, although I take it from your feedback that both the backwards and forwards passes are sparse, along with the mask update itself. This brings up an even bigger problem in my mind however, that is if both the backwards and forward pass are sparse, even at mask update time, how does the method ever consider the contributions of other masked out weights? As the other reviewer you refer to highlights, this is from restricting the mask search space? Doesn't this remove any potential benefit of the bi-level optimization as compared to existing heuristic methods?
> >
> > Q2: Unfortunately while I'd be happy to see this addressed in the paper, this response doesn't really address my question now during the review period... was the paper updated? Is there anything else I should be referring to to understand the impact of this better?
> >
> > Q3: This does address my question on the change in training time/steps for the models when being compared. Please do explain this in the paper also.

---

> ### Author Response · Authors · 2023-11-20
>
> **This is the 2/2 part of the response.**
>
> Below are our responses about the left questions.
>
> Q4. Significance of ImageNet Results and Variance
>
> A4. The reason that we report std dev for CIFAR and not ImageNet is because the data is meaningful for CIFAR but not quite informative for ImageNet. We understand the importance of providing a comprehensive understanding of the results. We will include the variance across the three runs of the ImageNet experiments in the paper to better convey the robustness and significance of the reported results.
>
>
> Q5. Regarding Mask Development with IoU
>
> A5. We appreciate the feedback. We justify the IoU changes for better mask development by providing the final accuracy result. It shows in the Figure 4 that BiDST achieves the best accuracy compared to the baseline methods (RigL, MEST) that uses manually set mask update policy. From the IoU curves, we can see that the purpose for other baselines that use manually set update policy is to ensure a reasonably large search ratio on the mask (e.g., MEST uses different levels of search ratio, while RigL uses a cosine annealing schedule for mask update ratio). However, such heuristic settings cannot always ensure the mask searching process is naturally suitable for the network. Additionally, a similar approach that uses IoU to analyze the accuracy can be found in [R2]. So we believe it is a solid metric. We will provide a more nuanced explanation in section 3.3. The revised section will explicitly justify the use of IoU and clarify the basis for claiming superior mask development by BiDST.
>
>
> Q6. Compiler Optimization and Sparse Training Speedups
>
> A6. We have explained in section 2.4, that the end-to-end on-device training overhead is because of the “recompilation” of the “static machine code”. The reviewer is correct about the execution code for sparse neural network is in the form of static code. However, DST needs to change this static code when the mask is updated (i.e., new computation graph). This recompilation needs time, which becomes the overhead time that slows down the overall training time. The benefit of using compiler code generation is because the execution efficiency is very high, even with the unstructured sparsity. According to the discussion and experiments in MEST, we can see that MEST successfully incorporates different sparsity in the compiler design, which certifies the possibility of using compiler optimization to accelerate BiDST. We use the CSR format to store the non-zero weights and integrate the compact form into the static code to achieve sparse acceleration. However, we must point out that the focus of this paper is in the algorithm-level innovation that not only achieves good DST accuracy with an analytical solution, but also helps to reduce the overhead related to hardware-level implementation.
>
> Q7. Background Placement and Coverage
>
> A7. We appreciate the reviewer’s opinion. We will re-organized the background section after the introduction section.
>
> [R2] Nowak, Aleksandra I., et al. "Fantastic Weights and How to Find Them: Where to Prune in Dynamic Sparse Training." NeurIPS 2023.

---

> > ### Comment · Reviewer_MEAC · 2023-11-22
> >
> > Q4: Again, this is good to update in the paper, but I don't see an answer to my question on what the variance/std dev is on the ImageNet results. So what is the stddev/variance of the ImageNet runs, and are the results still significant in this light?
> >
> > Q5: If the purpose of this analysis is to compare the breadth of the mask search for the different DST methods, wouldn't an established metric for this, i.e. ITOP, make much more sense? I still don't understand why IOU is a good metric for this given your explanation.
> >
> > Q6: You present real-world timings, and those timings show significant acceleration for unstructured sparsity - so regardless of your claims on the motivation of the paper, the timings themselves must make sense/be explained. Using CSR is a common method for unstructured sparsity that we know does not show as significant an increasing in real-world timings with any existing libraries I'm aware of. I still don't understand what "We use the CSR format to store the non-zero weights and integrate the compact form into the static code to achieve sparse acceleration" means given this. Frankly these results are too good to be true compared to the existing literature, and although this doesn't mean you haven't achieved it, it does mean that the burden of proof is more than a vaguely worded sentence on this. If you had shared code I would be happy to dig through that to try and understand it better, but unfortunately it seems there is no code.

---

### Official Review · Reviewer_xhao · 2023-11-01

**Soundness:** 2 fair
**Presentation:** 2 fair
**Contribution:** 2 fair
**Rating:** 5
**Confidence:** 3

**Summary:**

The paper discusses a class of methods known as dynamic sparse training (DST). These are methods for efficiently training sparse neural networks. The paper draws connections between DST and bi-level optimization (BLO) and explains that BLO is a suitable framework for the two levels of mask and weight training in sparse learning. They evaluate a particular instantiation of this framework which iterates between applying a gradient update to the mask variables and training the masked weights with SGD.

**Strengths:**

The authors explain in detail how sparse training and bi-level optimization are related.

The method performs well, and the authors propose a method of restricting the mask search space in order to keep training efficient.

**Weaknesses:**

L2 regularization is generally applicable to other DST methods and should be tested. It appears BiDST with the smallest L2 penalty in the appendix would underperform relative to the other baselines.

For Figure 4, I do think it is interesting to point out that BiDST changes the mask more than the baselines do, but I'm not convinced that a quickly changing mask (Figure 4) is necessarily something we want.

I am also wondering why Figure 4 does not also reflect the frequency of the mask changes. In the tables it shows that BiDST usually makes fewer mask updates, but looking at Figure 4 I would assume all methods make one mask update per epoch.

**Questions:**

See above.

---

> ### Author Response · Authors · 2023-11-20
>
> Thank you for your thoughtful review, we will definitely reflect them in the new version of the paper.
>
> Q1. About the l2 regularization.
>
> A1. L2 regularization is widely used in deep model training to stabilize the training performance by decaying the weights to avoid overfitting. It is a training strategy and it is not quite relevant to the dynamic sparse training scheme. As far as we have investigated, there is no DST literature that focuses on the L2 regularization in their algorithm. And more importantly, our setting on L2 regularization is following MEST, which are very common settings and used widely. Thus, our settings are not with the smallest L2 and certainly would not underperform other baselines. Please see Table A.1 for detailed L2 regularization settings. It's essential to note that the focus of this paper is on introducing a novel bi-level optimization approach in the form of BiDST. We agree that comparing L2 regularization across different DST methods could be interesting for future work. However, the omission of specific regularization in BiDST is a deliberate choice to emphasize the core concept of bi-level optimization. The objective is to demonstrate the efficacy of simultaneously optimizing both weight and mask objectives, which is a distinct feature of BiDST. Including L2 regularization might dilute this emphasis, and we believe it warrants a separate investigation.
>
> Q2. Mask changing fast in Figure 4.
>
> A2. We appreciate the reviewer's observation. The dynamism in the mask changes, as shown in Figure 4, is indeed a notable characteristic of BiDST. The rapid changes signify the model's adaptability, exploring different sparse patterns during training. It's essential to clarify that the speed of mask changes alone may not be a conclusive metric; the quality and effectiveness of the changes matter significantly. We want to stress that the mask changing policy is of great importance to the final results. For example, RigL changes mask every 100 iterations, which is a very fast change. It uses a very large batch size and learning rate to push the mask to be changing fast thus accommodating a greedy search scheme. BiDST aims to efficiently explore the mask searching space, resulting in superior sparse patterns and higher accuracy. The dynamism reflects the model's ability to adapt quickly to different patterns, which can be advantageous for capturing complex relationships within the data.
>
> Q3. Mask update frequency for other methods in Figure 4.
>
> A3. According to the description in Section 3.3, we draw the IoU using **fixed intervals**, not set other baselines with the same update frequency. All other baselines are using their original update frequency settings. We just measure IoU based on two masks between the same training epochs. For example, RigL updates the mask every 100 iterations. To draw Figure 4, we set the fixed interval to every 5 epochs, which is about 20 times mask updates. We will use the new mask after 20 times of updates and compute IoU with the one mask at the beginning of the 5 epochs of training. We will add more explanations to our revised paper.
>
>
> [R1] Yuan, Geng, et al. "MEST: Accurate and fast memory-economic sparse training framework on the edge." NeurIPS 2021.

---

### Meta-Review · Area_Chair_Ur98 · 2023-12-11

**Metareview:**

While the reviewers appreciated some of the contributions of the papers, they all recommended rejection, thinking that it is not ready yet for publication. Reviewer MEAC raised some issues with the explanations in the paper which were not properly addressed by the authors. The authors are encouraged to take their detailed feedback in consideration to make a major revision.

**Justification For Why Not Higher Score:**

The paper is not ready for publication at ICLR.

**Justification For Why Not Lower Score:**

N/A

---

### Decision · Program_Chairs · 2024-01-16

Reject